# Knowledge, attitude and practice towards complementary and alternative medicine and associated factors among health care professionals in public health facilities of Hadiya Zone, South Ethiopia

**Belay Erchafo** [ID]*, **Lonsako Abute, Tegegn Tedesse, Tagesse Sedoro, Aregash Mecha**

Department of Public Health, College of Medicine and Health Sciences, Wachemo University, Hosaena, Ethiopia

* erchafobelay@gmail.com

**Data Availability Statement:** All relevant data are within the paper and its Supporting Information files.

## Abstract

### Background

Even if modern medicine is becoming more widely available, a considerable portion of the world's population continues to rely on complementary and alternative medicine. Complementary and alternative medicine is used by 80% of the population in developing countries for their health care. The study was conducted to assess Knowledge, Attitude and practice towards complementary and alternative medicine and associated factors among health care professionals in public health facilities of Hadiya Zone, South Ethiopia.

### Methods

The study was conducted in Hadiya Zone from October 10 to October 30, 2019. A facility-based cross-sectional study design was employed using simple random sampling methods. Three hundred sixty six health care professionals were selected using a simple random sample method. The data were collected using a standardized self-administered and pre-tested questionnaire that was adapted from different literatures. We used descriptive statistics, as well as bivariate and multivariate logistic regression analysis. An association was determined using P-values less than 0.05 and 95 percent confidence intervals. The results of the study were presented using texts and tables.

### Results

Three hundred and fifty-five respondents were interviewed, with a 97 percent respondent rate. Two hundred thirty-two (65.4%) of the study participants have good knowledge towards complementary and alternative medicine, 216 (60.8%) have a favorable attitude toward complementary and alternative medicine, and 182 (51.3%) have utilized it in the previous two years. Some of the factors associated with knowledge, attitude, and practice towards complementary and alternative medicine were health care professionals who have

**Funding:** This study was funded by Wachemo University through a staff research grant with grant reference number RCSD/09/2013. The funders had no role in study design, data collection and analysis, the decision to publish, or the preparation of the manuscript.

**Competing interests:** The authors have declared that no competing interests exist.

contact with patients who seek complementary and alternative medicine [AOR = 1.89(95% CI; 1.18, 3.03), female health care professionals [AOR = 2.43(95% CI; 2.68, 9.74), and more than six years work experience [AOR = 1.68(95% CI; 1.04, 2.71).

## Conclusion

The knowledge, attitude, and practice of complementary and alternative medicine among health care professionals were low. Creating communication lines with patients and facilitating the integration of complementary and alternative medicine with modern medicine.

## Introduction

Even if modern medicine is becoming more widely available, a considerable portion of the world's population continues to rely on complementary and alternative medicine (CAM). In many underdeveloped nations, CAM is used by up to 80% of the population. As is obvious, the majority of society in developing nations rely on CAM for a variety of reasons, including accessibility, price, and a perceived assumption that it is safer and more effective than modern medicine. As a result, health-care executives have been focusing on integration and looking for the positive elements of CAM [1–3]. Despite the widespread use of CAM by the global population, there is a significant gap in health care professionals' knowledge, attitude, and practice (KAP] toward CAM [4–7].

Eighty percent of Ethiopians rely on traditional medicine to maintain their health [8]. However, a concerted effort is needed to fully exploit the CAM potential for the benefit of the health-care system. Nonetheless, the National Health Policy of 1993 advocated that traditional medicines be gradually integrated with modern medicine by paying attention to traditional practices and recognizing the positive and negative elements through investigation and research [9–11].

The importance of CAM in Ethiopian health care is significant; in reality, knowledge of the scope and features of CAM practices is limited, resulting in CAM being a serious public health issue [12,13]. In recent decades, little has been done in Ethiopia to strengthen and promote the positive aspects of CAM, including relevant research to examine the prospects for its gradual integration into contemporary medicine [14]. Ethiopia's national drug strategy aims to merge traditional and contemporary medicine, and the Ministry of Health is developing a five-year road plan to achieve this goal. We believe that analyzing the KAP of health care providers toward CAM is required to achieve the strategic plan and goal of national drug policy.

In order to achieve universal health coverage, researchers must address the relationship between health care professionals' characteristics and their KAP on CAM [15,16]. Knowing about CAM might help people have a more positive attitude toward it, which can make it easier to include it in medical and health science education curriculum. In addition, evaluating health care workers' KAP of CAM can aid in its integration into the larger health-care system. As per researchers' knowledge of KAP regarding CAM among health care professionals, which has yet to be examined in the Ethiopian health system, there are large numbers of CAM users in the health care industry.

The findings of this study will be used by the Zonal Health Department to develop an action plan and will benefit a variety of stakeholders, including public health practitioners, program planners, and decision-makers, in order to promote the use of CAM and to develop appropriate policies and recommendations. Researchers in the field who are interested in the subject

can utilize the study's findings as a reference point. As a result, the objective of this study was to assess health care professionals KAP on CAM and associated factors in public health facilities in Hadiya Zone, South Ethiopia.

## Methods and materials

### Study area and study period

The study was conducted out in public health facilities in Hadiya zone, southern Ethiopia, from October 10 to October 30, 2019. Hadiya zone is part of the SNNP regional state, which is 194 kilometers south of Hawass, the southern regional state's city, and 230 kilometers southwest of Addis Ababa, Ethiopia's capital. The zone's projected population for 2016/2017 was roughly 1,573,841, according to data obtained from the zonal health department. A total of 54,455 people are estimated to be pregnant. The Zone has one specialized hospital, three primary hospitals, 61 health centers, and 305 health posts, each with two health extension workers per kebele (small administrative unit).

### Study design and participants

An institution-based cross-sectional study design was used by the quantitative data collection method. All health care professionals who work in Hadiya Zone public health facilities were considered as the source population, and sampled health care professionals in the randomly selected public health facilities were taken as the study population. Health care professionals who are critically ill and unable to communicate were not allowed to participate

### Sample size and sampling techniques

The sample size was calculated using a single population proportion formula with the following assumptions: Z: 95% confidence interval, P = proportion of health care professionals who practice alternative and complementary medicine (who use for personal use and recommend for others) (50 percent), a non-response rate of 10%. Because the source population was 3818, which was less than 10,000, a finite population correction was required, resulting in a sample size of 366. Simple random sampling was used to choose 19 health centers and two primary hospitals from a total of 62 health centers and three primary hospitals in Hadiya zone. The overall sample size was proportionally assigned for selected public health facilities in Hadiya Zone, Southern Ethiopia, depending on the number of health care professionals in their respective health facilities. Three hundred and sixty-six providers were selected from the public health facilities using a simple random sampling technique. During the data collection period, individual participants in each of the health facilities were selected by simple random sampling until the required sample size at each health facility was achieved.

### Data collection tools and measurement

Data was collected using a standardized self-administered questionnaire. The questionnaire was divided into four sections. The first section contained socio-demographic information (age, gender, specialization, qualifications). The CAM approaches are covered in the second section. The final section focuses on KAP's approach to CAM. Yes/no questions were utilized in the fourth section to address several issues surrounding integrative medicine, such as its applicability and training and teaching opportunities. Knowledge towards CAM was assessed with seven questions. Good knowledge: those health care professionals who scored equal to or above the mean score on seven knowledge questions. Poor knowledge: those health care professionals who scored below the mean score on seven knowledge questions. Attitude towards

CAM was assessed by ten questions put on likert's scale. Negative attitude: health care professionals who scored lower than the mean on ten attitude questions. Positive attitude: health care professionals who scored equal to or above the mean score on ten attitude questions. The practice was assessed by yes or no questions towards CAM.

### Ethical approval and consent to participation

The study was conducted after securing ethical approval from ethical review committee of Wachemo University, college of medicine and health sciences. All the participants were well informed about the purpose of the study, benefits and risks associated with the study, written consent was secured from each study participant before collecting the data. The participants were also informed that their responses would be kept confidential and their names would not be mentioned.

### Data processing and analysis

For analysis, data was entered into EPI data version 3.1 and then transferred to SPSS version 22. For categorical data, proportions were used to summarize participant characteristics. KAP level on complementary and alternative medicine and factors associated with KAP on complementary and alternative medicine, were assessed using bivariate and multivariable logistic regression analysis. In bivariate analysis, all significant variables with a p-value of less than 0.25 were considered candidates for multivariable logistic regression. Significant associations with dependent variables were defined as those with a p-value less than 0.05 in multivariable logistic regression. The statistical significance and level of KAP on CAM were assessed using an odds ratio with a 95 percent confidence interval and a p-value of 0.05. A self-administered questionnaire was used to collect data. The questionnaire was written in the English language. A pretest was conducted outside the study area with 10% of the sample size. Eight diploma holders were selected to collect data, while four public health officers with bachelor's degrees were assigned as supervisors. A day-long training session about the study's objectives and advantages, as well as persons' rights and informed consent, was given to data collectors and supervisors. On a daily basis, the consistency and completeness of the data were reviewed.

## Results

### Socio-demographic characteristics

A standardized questionnaire was used to interview 355 respondents, resulting in a 97 percent response rate. Two hundred thirty-one (65.1%) had a bachelor's degree. Two hundred twenty-two (62.5%) of the study participants have worked for six years or more. Two hundred and twenty-eight (64.2%) of the participants were male and two hundred thirty-two (65.4%) were married (Table 1).

### Knowledge, attitude, and practice of the health care professionals towards complementary and alternative medicine

Three hundred thirty-six (94.6%) of health care professionals were heard about complementary and alternative medicine. One hundred twenty-four percent of health care providers (34.9%) get information about complementary and alternative medicine from family, friends, and relatives, while one hundred fifty-six percent (43.9%) use vitamins and other nutritional therapies from various complementary and alternative medicine practices. Due to cost, 185 (52.1%) of study participants prefer complementary and alternative medicine to modern medicine, and 207 (58.3%) of health care professionals recommend CAM for their patients. One

**Table 1. Socio demographic characteristics of health care professionals in selected primary health care unit in Hadiya Zone, Southern Ethiopia, April, 2021 (n = 335).**

| Variables | Categories | Frequency | Percent |
|---|---|---|---|
| Educational level | Diploma | 124 | 34.9 |
| | Degree | 231 | 65.1 |
| Work experience of health care professionals | 1–5 years | 129 | 36.3 |
| | 6 years and above | 226 | 63.7 |
| Age of participants | 20–29 | 232 | 65.4 |
| | 30–39 | 123 | 34.6 |
| Type of health facilities | Primary hospital | 120 | 33.8 |
| | Health center | 235 | 66.2 |
| Sex of the participants | Female | 127 | 35.8 |
| | Male | 228 | 64.2 |
| Field of the study | Nurse | 116 | 32.7 |
| | Health officer | 143 | 40.3 |
| | GP | 31 | 8.7 |
| | Midwifery | 26 | 7.3 |
| | Pharmacy | 39 | 11 |
| Marital status of the study participants | Single | 117 | 32.9 |
| | Married | 232 | 65.4 |
| | Divorced | 6 | 1.7 |

hundred eighty-six (52.4%) of the health care professionals had regular interaction with patients seeking CAM. Two hundred ninety-eight (83.9%) of health-care professionals are aware of the harmful effects of complementary and alternative medicine.

## The prevalence of knowledge, attitude, and practice towards complementary and alternative medicine among health care professionals

About 232 (65.4%) of the health care professionals have good knowledge of complementary and alternative medicine, 216 (60.8%) have a positive attitude toward complementary and alternative medicine, and 182 (51.3%) of the study participants have used complementary and alternative medicine in the previous two years (Table 2).

## Factors associated with knowledge of health care professionals towards complementary and alternative medicine

Educational status had a p-value of 0.015, health facility had a p-value of 0.012, study participants' age had a p-value of 0.011, study participants' sex had a p-value of 0.003, work

**Table 2. Knowledge, attitude, and practice of the health care professionals towards complementary and alternative medicine in selected primary health care unit in Hadiya Zone, Southern Ethiopia, April, 2021 (n = 335).**

| Variables | Categories | frequency | percent |
|---|---|---|---|
| Participants knowledge level | Good knowledge | 232 | 65.4 |
| | Poor knowledge | 123 | 34.6 |
| Attitude of participants | Positive attitude | 216 | 60.8 |
| | Negative attitude | 139 | 39.2 |
| Practice of participants | Yes | 182 | 51.3 |
| | No | 173 | 48.7 |

experience had a p-value of 0.002, and contact with patients seeking CAM had a p-value of 0.035 in bivariate analysis (Table 3).

## Factors associated with attitude of health care professionals towards complementary and alternative medicine

Educational status had a p-value of 0.017, health facility had a p-value of 0.022, study participants' age had a p-value of 0.037, study participants' sex had a p-value of 0.002, work experience had a p-value of 0.58, and contact with patients seeking CAM had a p-value of 0.001 in bivariate analysis (Table 4).

## Factors associated with practice of health care professionals towards complementary and alternative medicine

Variables like health facility had a p-value of 0.135, sex of study participants had a p-value of 0.081, and contact with patients seeking CAM had a p-value of 0.001 in bivariate analysis (Table 5).

# Discussion
## Knowledge towards complementary and alternative medicine

The high demand towards CAM by community and health care professionals to combat disease and to ensure universal health care coverage through primary health care approach requires great emphasis and effort to integrate into conventional medicine [17,18].

**Table 3. Bivariate and multivariable analysis of factors associated with knowledge towards complementary and alternative medicine among health care professionals in selected primary health care unit in Hadiya Zone, Southern Ethiopia, April, 2021 (n = 355).**

| Variables | Knowledge | | p-value and OR(95% CI) | | | |
|---|---|---|---|---|---|---|
| | Good | Poor | p-value | COR(95%CI) | p-value | AOR(95%CI) |
| Contact with patients who seek CAM | | | | | | |
| Yes | 131 | 55 | 0.035* | 1.61(1.03,2.43) | 0.008 | 1.89(1.18,3.03) |
| No | 101 | 68 | | 1 | | 1 |
| Sex | | | | | | |
| Female | 104 | 23 | 0.000* | 3.53(2.09,5.95) | 0.004 | 2.43(2.68,9.74) |
| Male | 128 | 100 | | 1 | | 1 |
| Work experience | | | | | | |
| 6 years and above | 161 | 65 | 0.002* | 2.02(1.28,3.17) | 0.034 | 1.68(1.04,2.71) |
| Below 6 years | 71 | 58 | | 1 | | 1 |
| Age of the study participants | | | | | | |
| 30–39 | 97 | 26 | 0.012 | 2.68(1.62,4.44) | 0.25 | 0.011(0.00,0.57) |
| 20–29 | 135 | 97 | | 1 | | 1 |
| Health facility | | | | | | |
| Primary hospital | 96 | 24 | 0.001 | 2.92(1.74,4.88) | 0.2 | 5.15(0.72,36.5) |
| Health center | 136 | 99 | | 1 | | 1 |
| Educational status | | | | | | |
| Diploma | 98 | 26 | 0.031 | 2.72(1.64,4.52) | 0.51 | 2.48(0.17,36) |
| Degree | 134 | 97 | | 1 | | 1 |

Note: * Statistically significant at p-value < 0.05, OR (95% CI).

**Table 4. Bivariate and multivariable analysis of factors associated with attitude towards complementary and alternative medicine among health care professionals in selected primary health care unit in Hadiya Zone, Southern Ethiopia, April, 2021 (n = 355).**

| Variables | Attitude | | p-value and OR(95% CI) | | | |
|---|---|---|---|---|---|---|
| | Positive | Negative | p-value | COR(95%CI) | p-value | AOR(95%CI) |
| Contact with patients who seek CAM | | | | | | |
| Yes | 136 | 50 | **0.000*** | **0.33(0.21,0.52)** | **0.000** | **3.6(2.27,5.8)** |
| No | 80 | 89 | | 1 | | 1 |
| Sex | | | | | | |
| Female | 91 | 36 | **0.002*** | **0.48(0.3,0.76)** | **0.003** | **2.72(2.85,9.74)** |
| Male | 125 | 103 | | 1 | | 1 |
| Work experience | | | | | | |
| 6 years and above | 148 | 78 | 0.18 | 0.58(0.38,0.92) | 0.65 | 1.58(0.97,2.55) |
| Below 6 years | 68 | 61 | | 1 | | 1 |
| Age of the study participants | | | | | | |
| 30–39 | 84 | 39 | 0.037 | 0.62(0.38,0.98) | 0.43 | 0.011(0.00,0.57) |
| 20–29 | 132 | 100 | | 1 | | 1 |
| Health facility | | | | | | |
| Primary hospital | 83 | 37 | 0.022 | 1.72(1.08,2.74) | 0.28 | 2.9(0.49,21.3) |
| Health center | 133 | 102 | | 1 | | 1 |
| Educational status | | | | | | |
| Diploma | 86 | 38 | 0.017 | 0.57(0.36,0.9) | 0.51 | 1.48(0.17,3) |
| Degree | 130 | 101 | | 1 | | 1 |

Note: * Statistically significant at p-value < 0.05, OR (95% CI).

In this study 232 (65.4%) study participants had good knowledge of complementary and alternative medicine. In another study conducted 51.65% of the study participants were found to have good knowledge about CAM. In the study conducted in Doha, Qatar about 60.9% of the study participants had good knowledge of CAM. In a study conducted in Turkey through a cross-sectional, the knowledge levels of health care workers towards CAM were adequate. Participants in this study showed a higher level of knowledge about CAM than in previous investigations. In this study, 336 (94.6%) of health-care professionals were informed about CAM. Vitamins and other nutritional therapies (156, 43.9%), medical herbalism (90, 25.4%), massage

**Table 5. Bivariate and multivariable analysis of factors associated with practice towards complementary and alternative medicine among health care professionals in selected primary health care unit in Hadiya Zone, Southern Ethiopia, April, 2021 (n = 355).**

| Variables | Practice | | p-value and OR(95% CI) | | | |
|---|---|---|---|---|---|---|
| | Yes | No | p-value | COR(95%CI) | p-value | AOR(95%CI) |
| Contact with patients who seek CAM | | | | | | |
| Yes | 134 | 52 | **0.000*** | **6.49(4.09,10.32)** | **0.025** | **1.12(1.27,2.38)** |
| No | 48 | 121 | | 1 | | 1 |
| Sex | | | | | | |
| Female | 73 | 54 | 0.081 | 1.47(0.95,2.28) | 0.167 | 0.115(0.07,0.189) |
| Male | 109 | 119 | | 1 | | 1 |
| Health facility | | | | | | |
| Primary hospital | 65 | 55 | 0.135 | 1.19(0.76,1.85) | 0.16 | 0.183(0.016,2.058) |
| Health center | 117 | 118 | | 1 | | 1 |

Note: * Statistically significant at p-value < 0.05, OR (95% CI).

(40, 11.3%), spiritual/faith healing (32, 9%), and traditional bone setting (8, 2.3%) were the types of CAM that health care professionals were familiar with. Medicinal herbs (168, 55.7 percent), spiritual or faith healing (57, 18.8%), traditional bone setting (41, 13.7%), and massage (21, 7%) were all mentioned in another study on CAM conducted in Trinidad and Tobago. Different results were found in a study conducted in Doha, Qatar, which showed that health professionals were more aware of psychotherapy and counseling, diet and supplements, acupuncture and massage. The possible difference for this might be having so many information channels and the content of the medical curriculum [2,19–22].

In this study, 298 (83.9%) of health care professionals were aware of the negative consequences of complementary and alternative medicine, with diarrhea, vomiting, abdominal discomfort, and skin discoloration being the most common adverse effects recorded. More than half of health care professionals, 184 (60.79%), were aware of the principal negative effects of complementary and alternative medicine, such as diarrhea (111, 36.64%), vomiting (62, 20.5%), and abdominal discomfort (62, 20.5%), according to another study on herbal medicine research (45, 14.91%). In this study, a higher percentage of respondents were aware of the risks associated with complementary and alternative medicine. The fact that there are so many ways to acquire information could be the possible reason [17].

## Attitude of complementary and alternative medicine

In this study 216 (60.8%) health care professionals had a positive attitude towards CAM. In another study conducted in Doha, Qatar about 83.8% of health care professionals had a positive attitude towards CAM. The possible reason for this might be due to contextual differences regarding the social condition of the society and the health policy of the countries. More than half of the respondents in our study, 174 (57.62%), agreed that CAM should be integrated into modern medicine. In contrast, a research conducted in Doha, Qatar found that approximately 97.5% percent of the study participants expressed a strong desire to attend CAM courses. A growing number of people in the health-care industry are calling for the integration of CAM into modern treatment. When it came to the effectiveness of complementary and alternative medicine, 222 (73.5%) participants disagreed or strongly disagreed that it is more effective than modern medicine. A majority of health care professionals, 236 (78.2%), disagree that CAM is safer than modern medicine. The majority of respondents disagreed (133, 44%) or strongly disagreed (82, 27.2%) about visiting CAM practitioners over modern medicine practitioners as their first choice. Only 60 (19.8%) of participants said they would go to a CAM practitioner before going to a doctor. A total of 191 (63.4%) of the study participants felt that complementary and alternative treatment is less expensive than modern treatment [17,22].

## Practice related to complementary and alternative medicine

In this study, 182 (51.3%) of participants had used CAM in the previous two years. Another study in Trinidad and Tobago found that the majority of respondents (74.22%) had used CAM in the previous two years. Only 33% of health care professionals utilized CAM in a study conducted in Doha, Qatar. Time differences, various degrees of experience of health care professionals with CAM therapies, the presence of CAM facilities, and different levels of interest and concern about CAM among countries could all contribute to discrepancies between studies [21].

## Factors associated with KAP on complementary and alternative medicine

Health care professionals who had interaction with patients seeking CAM were 1.9 times more likely than those who had no contact with patients seeking CAM to have good knowledge of

CAM. Female health care professionals were 2.5 times more likely than male health care professionals to have good knowledge of CAM, and health care professionals with 6 years of work experience or more were 1.7 times more likely to have good knowledge of CAM than health care professionals with less than 6 years of work experience. Healthcare professionals who had contact with patients seeking CAM were 3.6 times more likely to have a positive attitude about CAM than those who had no interaction with patients seeking CAM. Female health care providers were 2.7 times more likely than male health care providers to have a favorable attitude toward CAM. Health care providers who had contact with patients seeking CAM were 1.2 times more likely to utilize CAM than those who had no contact with patients seeking CAM [22,23].

## Limitation of the study

The exclusion of critically ill health care professionals due to physiological problems is a fact; there is a high tendency to use CAM. This may underestimate the practice of CAM among health care professionals.

## Conclusion

Knowledge, attitude, and practice of complementary and alternative medicine among health care professionals is low in this study area, and sex of health care professionals, contact with patients seeking complementary and alternative medicine, and work experience were factors associated with knowledge, attitude, and practice of complementary and alternative medicine in public health facilities in Hadiya zone, Southern Ethiopia. We recommend Integration of the complementary and alternative medicine with modern medicine, including it into the curriculum, establishing a communication channel with patients, and conducting additional research on the efficacy and quality of complementary and alternative medicine

## Supporting information

**S1 Questionnaire.**
(DOCX)

## Acknowledgments

We would like to forward our gratitude to Wachemo University, study participants, data collectors and supervisors.

## Author Contributions

**Conceptualization:** Belay Erchafo.

**Data curation:** Belay Erchafo.

**Formal analysis:** Belay Erchafo.

**Investigation:** Belay Erchafo.

**Methodology:** Belay Erchafo, Lonsako Abute, Tegegn Tedesse, Tagesse Sedoro, Aregash Mecha.

**Project administration:** Belay Erchafo.

**Software:** Belay Erchafo, Lonsako Abute, Tegegn Tedesse, Tagesse Sedoro, Aregash Mecha.

**Supervision:** Belay Erchafo, Lonsako Abute, Tegegn Tedesse, Tagesse Sedoro, Aregash Mecha.

**Writing – original draft:** Belay Erchafo.

**Writing – review & editing:** Belay Erchafo, Lonsako Abute, Tegegn Tedesse, Tagesse Sedoro, Aregash Mecha.

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
