## [Decision Letter · Decision Letter 0]

11 Mar 2022

PONE-D-21-34689Knowledge, Attitude and practice towards complementary and alternative medicine among health care professionals in public health facilities of Hadiya zone, South EthiopiaPLOS ONE

Dear Dr. lubago,

Thank you for submitting your manuscript to PLOS ONE. After careful consideration, we feel that it has merit but does not fully meet PLOS ONE’s publication criteria as it currently stands. Therefore, we invite you to submit a revised version of the manuscript that addresses the points raised during the review process.

We look forward to receiving your revised manuscript.

Kind regards,

Jenny Wilkinson, PhD

Academic Editor

PLOS ONE

Journal Requirements:

a) Did participants provide their written or verbal informed consent to participate in this study?

We would like to forward our gratitude to Wachemo University for providing fund for this study. 

6. We note you have included a table to which you do not refer in the text of your manuscript. Please ensure that you refer to Table 3, 4 and 5 in your text; if accepted, production will need this reference to link the reader to the Table.

Additional Editor Comments:

Thank you for your submission. Reviewer comments are provided for your consideration; these focus on making the text of your manuscript clearer for readers.

Reviewers' comments:

Reviewer's Responses to Questions

**Comments to the Author**

1. Is the manuscript technically sound, and do the data support the conclusions?

Reviewer #1: Yes

Reviewer #2: Yes

2. Has the statistical analysis been performed appropriately and rigorously? 

Reviewer #1: Yes

Reviewer #2: Yes

3. Have the authors made all data underlying the findings in their manuscript fully available?

Reviewer #1: Yes

Reviewer #2: Yes

4. Is the manuscript presented in an intelligible fashion and written in standard English?

Reviewer #1: Yes

Reviewer #2: Yes

5. Review Comments to the Author

Reviewer #1: (1) Brief information about data collection tools and measurement can be given (page ii, line 26).

(2) Information about the reasons for using TAT can be given (page, 1 line 45).

(3) It would be helpful to use a source containing a recent research done on page 2, line 67-69 (Sarman, A., & Uzuntarla, Y. (2022). Attitudes of healthcare workers towards complementary and alternative medicine practices: A cross-sectional study in Turkey. European Journal of Integrative Medicine, 49, 102096. https://doi.org/10.1016/j.eujim.2021.102096. Source address: https://www.sciencedirect.com/science/article/abs/pii/S1876382021008143).

(4) Is there any particular reason for not including critically ill healthcare workers? Since those with such diseases tend to use CAM more, it can be written in detail in the limitation section of the study (page 3, line 98-99).

(5) The methodology of the research is stated clearly and comprehensibly. The population and sample selection is appropriate. I think there is no problem with the randomization method (page 3-4-5, line 84-156).

(6) The sum of the column percentages of the marital status of the study participants should be checked. It is stated in the table as 100.1% (page 7, line 167; Table 1)

(7) It would be helpful to use a source containing a recent research done on page 14, line 231-232 (Sarman, A., & Uzuntarla, Y. (2022). Attitudes of healthcare workers towards complementary and alternative medicine practices: A cross-sectional study in Turkey. European Journal of Integrative Medicine, 49, 102096. https://doi.org/10.1016/j.eujim.2021.102096. Source address: https://www.sciencedirect.com/science/article/abs/pii/S1876382021008143).

(8) The manuscript is presented in an intelligible fashion and written in standard English.

(9) I cannot advise on statistical analysis of the paper.

Reviewer #2: Comment #1

The title of the study should have to be changed in to “knowledge, Attitude and practice towards complementary and alternative medicine and associated factors among health care professionals in public health facilities of Hadiya zone, South Ethiopia” in stage of knowledge, Attitude and practice towards complementary and alternative medicine among health care professionals in public health facilities of Hadiya zone, South Ethiopia

Comment #

In abstract the objective of the study should be stated in clear and unambiguous way and it has to be similar to the title of the study; in this paper the objective of the study were mixed with significance of the study; and the author should be amend accordingly as follows;

Even if modern medicine is becoming more widely available, a considerable portion of the world's population continues to rely on complementary and alternative medicine. Complementary and alternative medicine is used by 80% of the population in developing countries for their health care. The study was conducted to assess Knowledge, Attitude and practice towards complementary and alternative medicine among health care professionals in public health facilities of Hadiya zone, South Ethiopia.

Comment #3

Methods

The methods part should be starting by stating the study area first and; the authors should have to be corrected as “The studies were conducted in Hadiya zone from October 10 to October 30, 2019. A facility-based cross-sectional study design was employed using simple random sampling methods. Three hundred sixty six health care professionals were selected using a simple random sample method. We used descriptive statistics, as well as bivariate and multivariate logistic regression analysis. An association was determined using P-values less than 0.05 and 95 percent confidence intervals. The results of the study were presented using texts, graphs and tables.

Comment #4, in result part of abstract

Results

Three hundred and fifty-five respondents were interviewed, with a 96 percent response respondent rate. Two hundred thirty two (65.4%) of the study participants have good knowledge towards complementary and alternative medicine, 216 (60.8%) have a favorable attitude toward complementary and alternative medicine, and 182 (51.3%) have utilized it in the previous two years. Some of the factors associated with knowledge, attitude, and practice towards complementary and alternative medicine were health care professionals who have contact with patients who seek complementary and alternative medicine [AOR = 1.89(95% CI; 1.18, 3.03), female health care professionals [AOR = 2.43(95% CI; 2.68, 9.74), and more than six years work experience [AOR = 1.68(95% CI; 1.04, 2.71).

Comment #5

In the background of the study in introduction part in first paragraph; in second sentence that said as a result, health46 care executives have been focusing on integration and looking for the positive elements of 47 CAM [1, 2, 3] in this way because it is already appear for the third times in the paragraph and this comment should be followed through ought the study.

Comment #6

In the last paragraph in the last sentence of background the sentence which states the objective of the study that were mentioned by authors is better if it is changed as “the objective of this study were to assess health care workers knowledge, attitude, and practice of complementary and alternative medicine and associated factors among in public health facilities in Hadiya zone, south Ethiopia”.

Comment #7

In methods and material of the study design and participants part the author’s should have re-write the idea that is mentioned in this subtitle because the study design and participants part should have to clearly described.

Comment #8

In Result part

Socio-demographic characteristics

A standardized questionnaire was used to interview rather than 350, it has to corrected as “355” respondents, resulting in a rather than 96 percent; has to be corrected as “97” percent response rate. Sixty-one percent of health-care providers should have to changed as “Two hundred thirty one (65.1%) had a bachelor's degree”. Two hundred twenty-two (62.5%) of the study participants have worked for six years 162 or more. Two hundred and twenty-eight (64.2%) of the participants in the study were male. Two 163 hundred thirty-two (65.4%) were married, and two hundred twenty-two (62.5%) were Hadiya by ethnic group, the sentence which describe about ethnicity should be deleted because it is not included in the table (Table 1).

6. PLOS authors have the option to publish the peer review history of their article (what does this mean?). If published, this will include your full peer review and any attached files.

Reviewer #1: No

Reviewer #2: **Yes: **Tariku Tesfaye

---

## [Author Response · Author response to Decision Letter 0]

17 May 2022

Academic editor: we addressed all the questions and concerns raised in the manuscript in a point-by-point fashion. They were very pertinent. 

Reviewer 1: we addressed all the questions and concerns raised in the manuscript in a point-by-point fashion. They were very pertinent.

Reviewer 2: we addressed all the questions and concerns raised in the manuscript in a point-by-point fashion. They were very pertinent.

---

## [Decision Letter · Decision Letter 1]

26 Aug 2022

Knowledge, attitude and practice towards complementary and alternative medicine and associated factors among health care professionals in public health facilities of Hadiya Zone, South Ethiopia

PONE-D-21-34689R1

Dear Dr. lubago,

We’re pleased to inform you that your manuscript has been judged scientifically suitable for publication and will be formally accepted for publication once it meets all outstanding technical requirements.

Kind regards,

Sergio A. Useche, Ph.D.

Academic Editor

PLOS ONE

Additional Editor Comments (optional):

The authors have addressed the comments provided by their referees in a good way. Therefore, the paper can be accepted for publication.

Reviewers' comments:

Reviewer's Responses to Questions

**Comments to the Author**

1. If the authors have adequately addressed your comments raised in a previous round of review and you feel that this manuscript is now acceptable for publication, you may indicate that here to bypass the “Comments to the Author” section, enter your conflict of interest statement in the “Confidential to Editor” section, and submit your "Accept" recommendation.

Reviewer #1: All comments have been addressed

2. Is the manuscript technically sound, and do the data support the conclusions?

Reviewer #1: Yes

3. Has the statistical analysis been performed appropriately and rigorously? 

Reviewer #1: Yes

4. Have the authors made all data underlying the findings in their manuscript fully available?

Reviewer #1: Yes

5. Is the manuscript presented in an intelligible fashion and written in standard English?

Reviewer #1: Yes

6. Review Comments to the Author

Reviewer #1: It seems that the article has become more appropriate after the revision. Thanks to the authors for careful revisions.

7. PLOS authors have the option to publish the peer review history of their article (what does this mean?). If published, this will include your full peer review and any attached files.

Reviewer #1: **Yes: **Abdullah SARMAN

---

## [Editor Report · Acceptance letter]

30 Aug 2022

PONE-D-21-34689R1 

Knowledge, attitude and practice towards complementary and alternative medicine and associated factors among health care professionals in public health facilities of Hadiya Zone, South Ethiopia 

Dear Dr. lubago:

I'm pleased to inform you that your manuscript has been deemed suitable for publication in PLOS ONE. Congratulations! Your manuscript is now with our production department. 

Kind regards, 

on behalf of

Dr. Sergio A. Useche 

Academic Editor

PLOS ONE